# AR phosphorylation and CHK2 kinase activity regulates IR-stabilized AR–CHK2 interaction and prostate cancer survival

Huy Q Ta[1†], Natalia Dworak[1†], Melissa L Ivey[1], Devin G Roller[1], Daniel Gioeli[1,2]*

[1]Department of Microbiology, Immunology, and Cancer Biology, University of Virginia, Charlottesville, United States; [2]Cancer Center Member, University of Virginia, Charlottesville, United States

**Abstract** We have previously demonstrated that checkpoint kinase 2 (CHK2) is a critical negative regulator of androgen receptor (AR) transcriptional activity, prostate cancer (PCa) cell growth, and androgen sensitivity. We have now uncovered that the AR directly interacts with CHK2 and ionizing radiation (IR) increases this interaction. This IR-induced increase in AR–CHK2 interactions requires AR phosphorylation and CHK2 kinase activity. PCa associated CHK2 mutants with impaired kinase activity reduced IR-induced AR–CHK2 interactions. The destabilization of AR – CHK2 interactions induced by CHK2 variants impairs CHK2 negative regulation of cell growth. CHK2 depletion increases transcription of DNAPK and RAD54, increases clonogenic survival, and increases resolution of DNA double strand breaks. The data support a model where CHK2 sequesters the AR through direct binding decreasing AR transcription and suppressing PCa cell growth. CHK2 mutation or loss of expression thereby leads to increased AR transcriptional activity and survival in response to DNA damage.

*For correspondence:
dgioeli@virginia.edu

†These authors contributed equally to this work

Competing interests: The authors declare that no competing interests exist.

## Introduction

Mammalian cells are continuously being bombarded by endogenous and exogenous forces that jeopardize the integrity of DNA. In response to DNA damage, a conserved network of signaling pathways known as the DNA damage response (DDR) is activated to maintain cell viability and genome stability (*Rhind and Russell, 2000*). Prostate cancer (PCa) remains one of the leading causes of death among men of all races (cdc.gov), as castration-resistant prostate cancer (CRPC) is currently incurable. Recently, the DDR has been a focus of PCa research since the androgen receptor (AR), a major driver of PCa, modulates the transcription of DDR genes and DNA repair (*Polkinghorn et al., 2013*; *Goodwin et al., 2013*; *Jividen et al., 2018*). We have previously shown that checkpoint kinase 2 (CHK2) negatively regulates androgen sensitivity and PCa cell growth (*Ta et al., 2015*).

CHK2 is a serine/threonine protein kinase that plays a crucial role in sensing DNA damage and initiating the DDR, which is comprised of cell cycle arrest, DNA repair, and apoptosis (*Matsuoka et al., 1998*). CHK2 consists of an amino-terminal SQ/TQ cluster domain (SCD) where threonine 68 serves as a substrate for phosphorylation by ataxia-telangectasia mutated (ATM) kinase *Kim et al., 1999*; a carboxy-terminal kinase domain (KD) and nuclear localization sequence *Ahn et al., 2004*; and a central forkhead-associated domain (FHA) that provides an interface for interactions with phosphorylated proteins (*Li et al., 2002*). Currently, there are approximately 24 CHK2 substrates in human cells that have been experimentally validated, including polo-like kinase 1 (PLK1), promyelocytic leukemia protein (PML), E2F1, p53, and cell division cycle 25C (CDC25C) (*García-Limones et al., 2016*). These studies show that one mechanism CHK2 utilizes to affect cellular function is through direct protein-protein interactions.

CHK2 association with PLK1 leads to its localization at centrosomes where it regulates mitotic entry (*Tsvetkov et al., 2003*). CHK2 autophosphorylation and activation are regulated by the tumor suppressor PML within nuclear matrix-associated structures called PML-nuclear bodies, which are nuclear matrix-associated structures (*Yang et al., 2002*). Binding to PML keeps CHK2 in an inactive state within these PML-nuclear bodies. In return, activated CHK2 can phosphorylate PML on S117 and induce PML-mediated apoptosis. CHK2 can also modify the transcription of apoptotic genes through direct binding and S364 phosphorylation of the E2F1 transcription factor in response to DNA damage, thereby stabilizing E2F1 and activating gene transcription (*Stevens et al., 2003*). CHK2 also regulates apoptosis through p53 phosphorylation, and promotion of p53-mediated cell death (*Hirao et al., 2000*). The interaction with the core domain of p53 induces an allosteric change in CHK2 which permits p53 S20 phosphorylation (*Bartek and Lukas, 2003*). Moreover, CHK2 modulates CDC25C localization by associating with and phosphorylating CDC25 on S216, which creates a binding site for 14-3-3 proteins (*Peng et al., 1997*). 14-3-3 proteins in turn sequester CDC25C in the cytoplasm and block the G2/M transition since cyclin dependent kinase 1 (CDK1) cannot be activated. Finally, our group has shown that CHK2 co-immunoprecipitated with AR in PCa cells and regulated growth, suggesting that AR may be a novel substrate of CHK2 (*Ta et al., 2015*). Thus, given the importance of CHK2 and AR to the DDR and prostate cancer growth, a full understanding of the functional consequences of the CHK2–AR interaction is required, with the hope of possible clinical applications towards CRPC.

Here, we uncovered novel molecular interactions between CHK2 and AR that provide mechanistic insight into our observation that CHK2 negatively regulates prostate cancer growth. We demonstrate that AR directly binds CHK2, and that this interaction increases with ionizing radiation (IR). The IR-induced increase in AR – CHK2 interaction requires AR phosphorylation on both S81 and S308 and CHK2 kinase activity, which is impaired in PCa associated CHK2 mutations. Furthermore, these CHK2 mutants exhibit diminished effects on prostate cancer cell growth. CHK2 depletion increases transcription of DNAPK and RAD54, clonogenic survival, and resolution of DNA double strand breaks (DSBs) following IR, while altering the kinetics of IR-induced γH2AX foci.

## Results

### AR directly binds CHK2

We previously showed that AR co-immunoprecipitated with CHK2 immune complexes in several prostate cancer cell lines (*Ta et al., 2015*). To determine whether this co-association was through direct protein-protein interaction, we performed Far western blotting (*Figure 1*, *Figure 1—figure supplement 1*; *Prickett et al., 2008*; *Wu et al., 2007*). We also used Far western blotting to test the impact of AR phosphorylation on CHK2 binding to AR. We focused on S81 and S308 phosphorylation since the phosphorylation of AR on S81 by CDK1 and CDK9 stimulates AR transcriptional activity and growth of PCa cells (*Chen et al., 2006*; *Gordon et al., 2010*; *Chen et al., 2012*). AR phosphorylation on S308 by CDK1 represses AR transcriptional activity and alters AR localization during mitosis (*Gioeli et al., 2002*; *Zong et al., 2007*). To generate purified protein for the far westerns, 293 T cells were transfected with mammalian plasmids expressing FLAG-wtAR, or the FLAG-AR mutants S81A, S81D, S308A, or S308D, FLAG-ERK2, or V5-wtCHK2 (*Figure 1A*). We used FLAG-ERK2 as a positive control since it has been reported that CHK2 physically associates with ERK1/2 in cancer cells (*Dai et al., 2011*). FLAG-ERK and FLAG-AR targets were immunoaffinity purified and resolved by SDS-PAGE. The target proteins (AR wt and phospho-mutants and ERK) on the membrane were probed with purified V5-wtCHK2 protein, crosslinked, and stained with V5 antibodies to detect bound V5-wtCHK2. Membranes were also immunoblotted with AR and ERK1/2 antibodies to confirm that the molecular weight of AR and ERK1/2 corresponded with the CHK2 signal, which then indicates direct protein-protein interaction. We found that V5-wtCHK2 bound to FLAG-wtAR, as well as the control FLAG-ERK2 (*Figure 1A*). Only AR S308A showed a statistically significant reduction in binding to CHK2 (adjusted p=0.035), although all AR phosphorylation mutants trended to decrease association with CHK2.

We observed the reciprocal binding of AR and CHK2 by Far western. Expression, purification, and Far western blotting were performed as described above using FLAG-wtCHK2 and FLAG-CHK2 mutants K373E or T387N as targets and HA-wtAR as a probe. This also showed that AR and CHK2

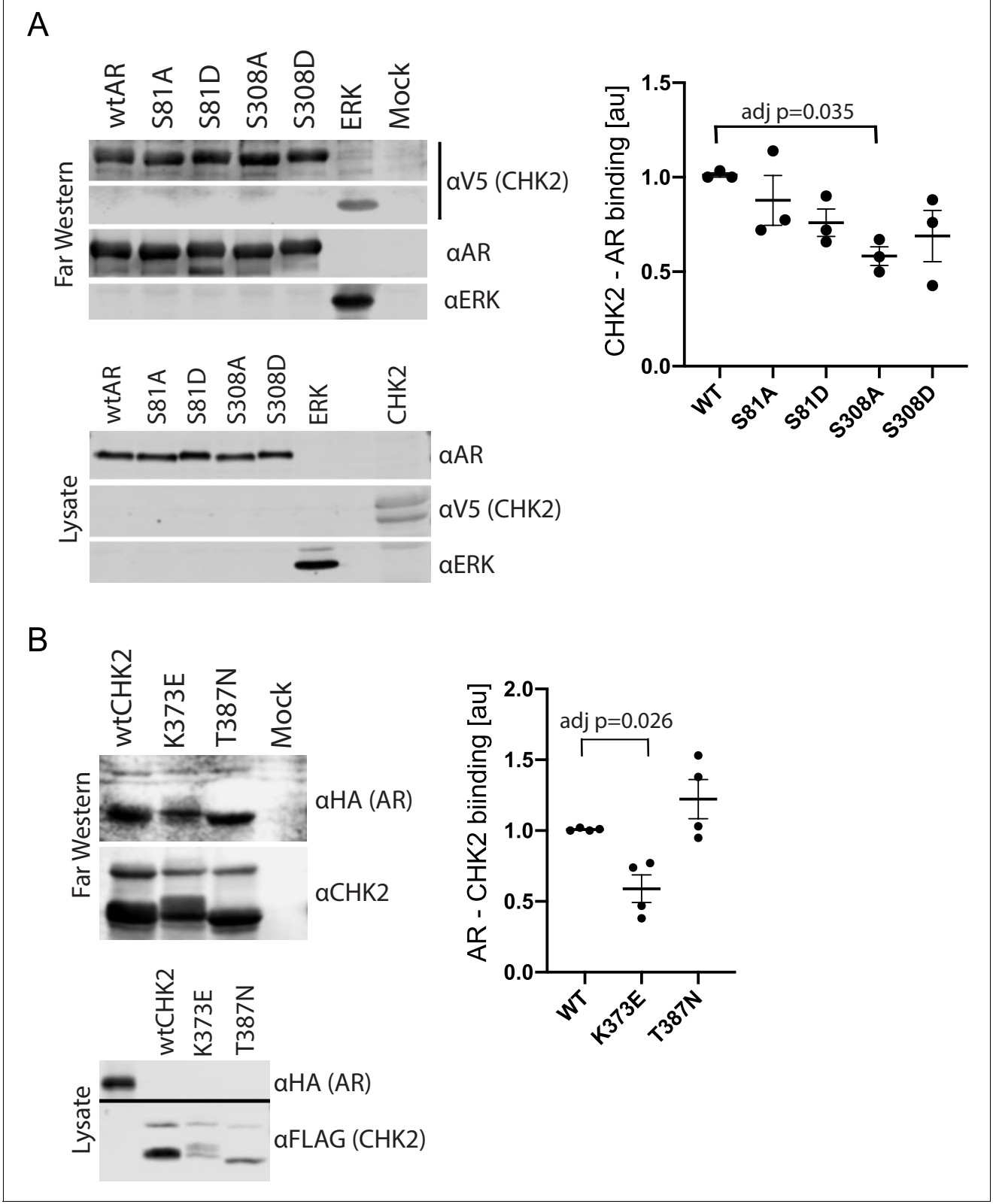

**Figure 1.** CHK2 directly binds AR. 293 T cells were transfected with Vector, FLAG-wtAR, or the FLAG-AR mutants S81A, S81D, S308A, or S308D, FLAG-wtCHK2, FLAG-CHK2 mutants K373E or T387N, FLAG-ERK2, HA-wtAR, or V5-wtCHK2 using Fugene 6. 48 hr following transfection, FLAG, HA, and V5 were immunoprecipitated, and far western blotting was performed. Membranes were blotted with the following antibodies: FLAG, HA, V5, AR, CHK2, and ERK2. Blots were visualized using the Odyssey CLx. (**A**) Probe = V5 wild-type CHK2; Targets = FLAG wtAR, or the FLAG-AR mutants S81A, S81D,

*Figure 1 continued on next page*

Figure 1 continued

S308A, or S308D, and FLAG-ERK2. Representative blots are shown, n = 3. (B) Probe = HA wtAR; Targets = FLAG-wtCHK2, FLAG-CHK2 mutants K373E or T387N. Representative blots are shown, n = 4. For (A) and (B), graphs are of relative AR – CHK2 binding. Error bars, SEM. Statistical differences were tested for using ANOVA with Dunnett's multiple comparisons test.

The online version of this article includes the following figure supplement(s) for figure 1:

**Figure supplement 1.** Far western blot schematic.

directly bind (*Figure 1B*). In these experiments, we also determined whether the CHK2 mutants in PCa with reduced kinase activity bind to AR. The K373E CHK2 mutation impairs CHK2 function suppressing cell growth and promoting survival in response to IR, as a result of reduced kinase activity due to the disruption of CHK2 autophosphorylation (*Higashiguchi et al., 2016*). Less is known about the heterozygous missense mutation T387N, but it is reported to diminish kinase activity (*Wasielewski et al., 2009*). We observed reduced AR – CHK2 binding (adj p=0.026) with the CHK2 K373E mutant. No significant change was observed with the CHK2 T387N mutant. These data suggest that the AR directly binds CHK2 and that AR phosphorylation and CHK2 kinase activity may impact that binding.

## Radiation increases AR – CHK2 association

Since IR is a standard of care for patients with localized advanced PCa, and CHK2 is a mediator of the DDR, we wanted to assess the impact of IR on AR – CHK2 interactions. Moreover, we wanted to test the influence of AR phosphorylation and CHK2 mutation on any effect of IR on the AR – CHK2 interaction. LNCaP cells were transduced with lentiviral particles containing wtAR or AR mutants S81A or S308A (*Figure 2A*). The association of AR was analyzed from endogenous CHK2 immune complexes generated one hour after irradiation. There was a 4-fold increase in AR co-immunoprecipitating with CHK2 after IR treatment of cells expressing wtAR. However, neither S81A nor S308A increased in association with CHK2 upon IR treatment indicating that phosphorylation of AR on S81 and S308 is required for the IR-induced increase in the interaction between AR and CHK2. Interestingly, in contrast to the AR – CHK2 association by Far western, the basal level of CHK2 and AR co-immunoprecipitation (co-IP) was unaffected by AR phosphorylation.

The requirement for AR phosphorylation for the IR-induced increase in AR – CHK2 association led us to test if AR phosphorylation on S81 and S308 was increased in response to IR. AR was immunoprecipitated from irradiated cells, and phospho-S81 and phospho-S308 were measured by western blotting using phospho-specific antibodies to those sites (*Gordon et al., 2010*; *Koryakina et al., 2015*; *Figure 2—figure supplement 1*). There were no significant changes in S81 or S308 phosphorylation in response to IR in either cell line. Thus, these findings indicate that while the intensity of S81 and S308 phosphorylation does not markedly change with IR, S81 and S308 phosphorylation is necessary for AR – CHK2 interactions under certain conditions.

## CHK2 kinase activity is required for AR – CHK2 interaction

To further determine if CHK2 kinase activity was necessary for the AR – CHK2 association and what impact IR has on CHK2 mutant association with AR, we tested if CHK2 mutants that are found in PCa and have impaired kinase activity could interact with the AR by co-IP. We expressed FLAG-wtCHK2, FLAG-K373E, or FLAG-T387N in combination with HA-wtAR in LNCaP cells. FLAG-CHK2 immunoprecipitations were performed and HA-AR association was evaluated (*Figure 2B*). In response to IR there was a striking increase in CHK2 – AR co-association in cells expressing FLAG-wtCHK2 and HA-wtAR. However, the amount of IR-induced increase in the AR – CHK2 association in cells expressing either K373E or T387N was dramatically reduced. No change in basal AR – CHK2 co-IP was observed. Immunoblotting of cell lysates used in the co-IPs indicate that the differences in co-IPed proteins was not due to protein expression (*Figure 2C*). These data indicate that the kinase activity of CHK2 is required for the optimal induction of the CHK2 and AR interaction in response to IR, and that the CHK2 mutants associated with PCa have a diminished ability to interact with the AR in irradiated cells.

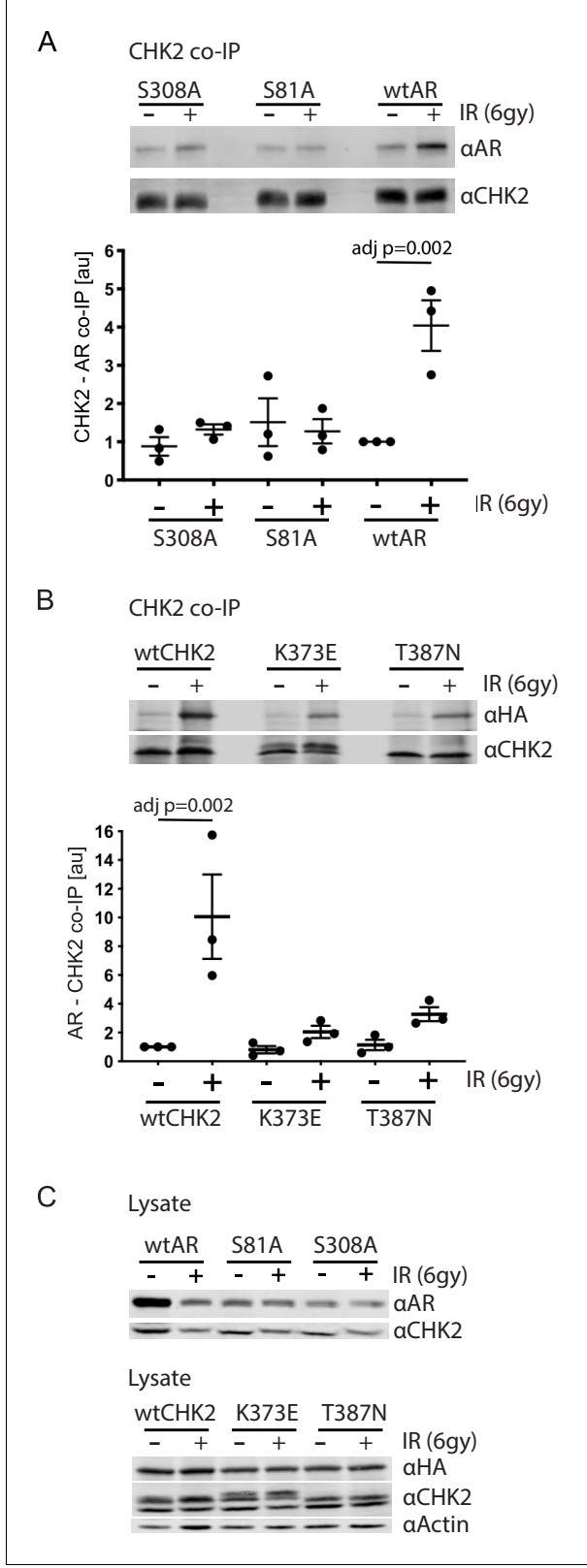

**Figure 2.** AR phosphorylation and CHK kinase activity regulates AR – CHK2 association. (**A**) AR – CHK2 interactions requires AR phosphorylation on S81 and S308. LNCaP cells were transduced with lentiviral particles expressing wtAR, S81A, or S308A for 48 hr. Cells were irradiated with 6Gy, and CHK2 was immunoprecipitated one hour after IR. Representative blots are shown. Blots were quantitated on Odyssey LICOR imaging system. Plotted

*Figure 2 continued on next page*

*Figure 2 continued*
is the AR signal normalized to total CHK2 and compared to untreated cells, n = 3. Error bars, SEM. Statistical differences were tested for using ANOVA with Sidak's multiple comparisons test. (**B**) Expression of CHK2 variants with reduced kinase activity inhibits the radiation-induced increase in AR – CHK2 interactions. LNCaP cells were transfected with HA-wtAR, HA-S81A, or HA-S308 in combination with FLAG-wtCHK2 for 48 hr using TransIT-2020 (Mirus). Cells were irradiated with 6Gy, and FLAG was immunoprecipitated using a magnetic bead system one hour after IR. Representative blots are shown. Plotted is the HA-AR signal normalized to total FLAG-CHK2 and compared to untreated cells. Error bars, SEM. Blots were quantitated on Odyssey LICOR imaging system. Statistical differences were tested for using ANOVA with Sidak's multiple comparisons test. (**C**) Control immunoblots of lysate from (**A**) and (**B**).

The online version of this article includes the following figure supplement(s) for figure 2:

**Figure supplement 1.** AR S81 and S308 phosphorylation are not altered with radiation.

## PCa associated CHK2 mutants limit suppression of PCa growth

Since CHK2 regulates the cell cycle and PCa cell growth (*Ta et al., 2015*; *Bartek and Lukas, 2003*), we investigated the effect of CHK2 mutations found in PCa, which disrupt the AR – CHK2 interaction, on CHK2 regulation of PCa cell growth (*Figure 3*). PCa cells were transduced with vector or shRNA specific to CHK2 to deplete endogenous CHK2, and then CHK2 expression was rescued with wtCHK2, K373E, or T387N. Cell growth was quantitated in the presence and absence of 0.05nM R1881 (synthetic androgen) seven days following transduction using CyQUANT to assess cell proliferation as a function of DNA content. In agreement with previous reports (*Ta et al., 2015*), hormone stimulated growth of vector-expressing cells and knockdown of CHK2 significantly augmented growth in all PCa cell lines tested (*Figure 3*). Here growth induced by CHK2 knockdown is likely an increase in cell proliferation, as the basal apoptotic rate in the PCa lines is low. Re-expression of wtCHK2, K373E, and T387N in CHK2-depleted cells markedly suppressed the increase in growth induced by CHK2 knockdown in all PCa cell lines tested. Interestingly, in LNCaP cells the extent of growth inhibition induced by K373E and T387N was significantly less than that generated by wtCHK2 (*Figure 3A*). The effect of CHK2 re-expression on cell growth in castration-resistant C4-2 (*Figure 3B*) and Rv1 (*Figure 3C*) cells was not significantly different between wtCHK2 and the kinase deficient K373E and T387N mutants, although the magnitude of inhibition caused by the mutants was consistently less than that produced by wtCHK2. Expression levels of wtCHK2, K373E, and T387N do not account for the difference observed between LNCaP and C4-2 or Rv1 (*Figure 3*) since the relative expression of wtCHK2, K373E, and T387N were similar across the cell lines. These observations suggest that CHK2 kinase activity limits the ability of CHK2 to negatively regulate PCa cell growth, especially in androgen dependent PCa.

## CHK2 suppresses IR induction of DNAPK and RAD54

Reports in the literature suggest that the AR is a critical regulator of genes in the DDR (*Polkinghorn et al., 2013*; *Goodwin et al., 2013*; *Jividen et al., 2018*) Therefore, we evaluated the impact of CHK2 knockdown on IR induced transcription of DDR genes in LNCaP cells (*Figure 4*). In our experiments, androgen or IR only modestly affected DNAPK and RAD54 transcript levels; we did not observe androgen or IR induction of XRCC2, XRCC3, XRCC4, XRCC5, MRE11, RAD51, FANC1 and BRCA1 transcripts as reported by others (data not shown) (*Polkinghorn et al., 2013*; *Goodwin et al., 2013*). This discrepancy is consistent with the observation that androgen regulation of DDR genes is specific to the model system, experimental conditions, and disease state examined (*Jividen et al., 2018*). Knockdown of CHK2 in LNCaP cells grown in CSS and stimulated with 1 nM DHT led to an increase in transcription of DNAPK and RAD54 in response to IR, suggesting that CHK2 may suppress AR transcription of DDR genes in response to IR.

## CHK2 effect on IR sensitivity and DNA repair

We next examined the impact of CHK2 interactions on cell survival and sensitivity to IR (*Figure 5A*). LNCaP cells expressing CHK2 (pLKO) or depleted of CHK2 (CHK2 KD) were exposed to increasing doses of IR, plated, and allowed to grow for 14 days. Clonogenic assays revealed that CHK2 knockdown promoted cell survival following ionizing radiation. This data, along with the data above

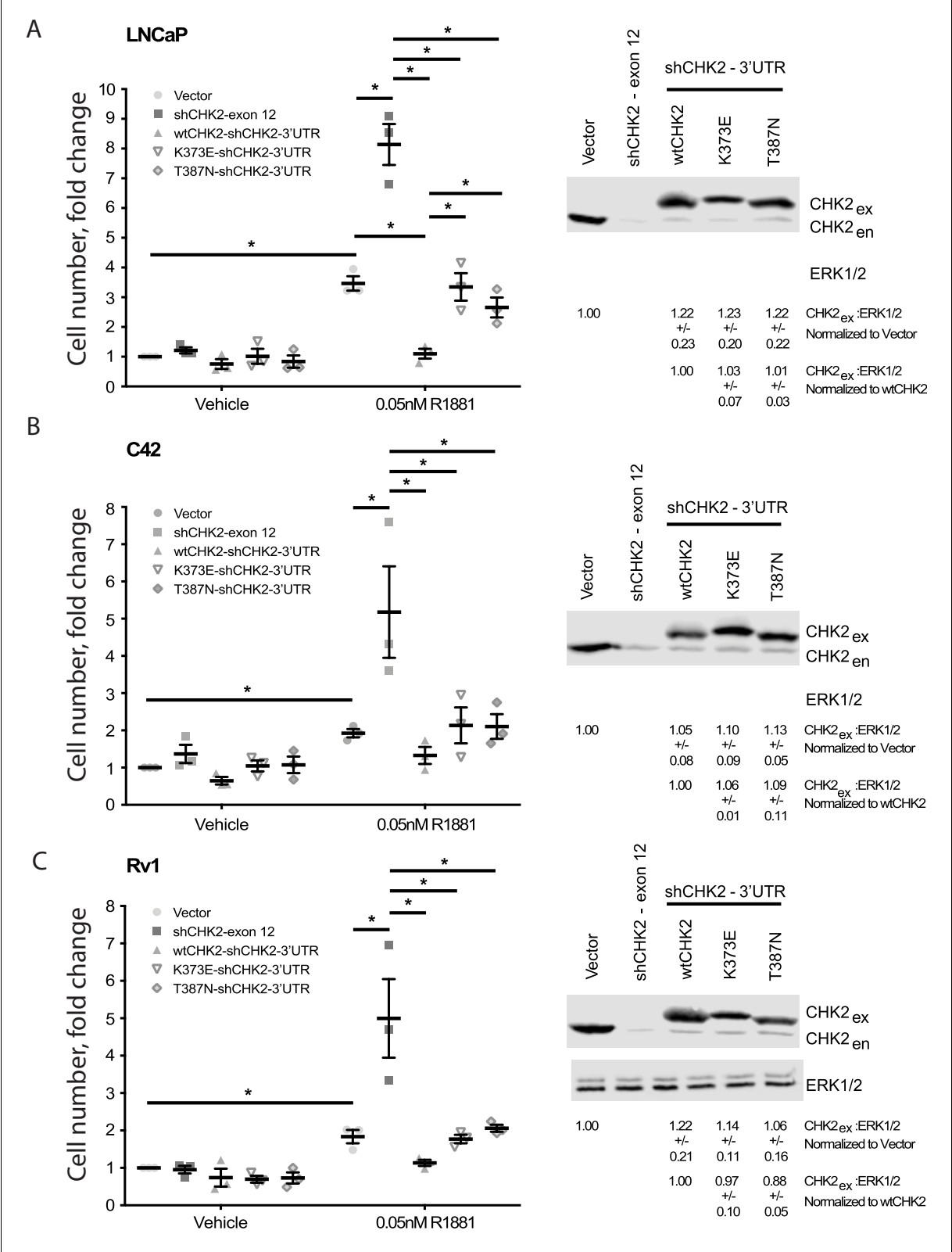

**Figure 3.** Wild-type CHK2 negatively regulates prostate cancer cell growth. (**A**) LNCaP, (**B**) C4-2, and (**C**) Rv-1 cells were transduced with lentiviral particles expressing vector, shCHK2-exon 12, or shCHK2-3'UTR in combination with wtCHK2, K373E, or T387N in the presence or absence of 0.05nM R1881. CyQuant assay was performed 7 days after transduction. Cell growth was compared with untreated vector control and the values were averaged
*Figure 3 continued on next page*

*Figure 3 continued*

across biological replicates. Error bars, SEM, n = 3. Statistical analysis was performed using ANOVA and Tukey test, *=p < 0.01. Representative blots of CHK2 expression are shown with ERK as a loading control.

The online version of this article includes the following figure supplement(s) for figure 3:

**Figure supplement 1.** Non-cropped images of western blots shown in the manuscript.

indicating that CHK2 suppresses AR transcription of DNA repair genes suggested the hypothesis that loss of CHK2 could facilitate DNA repair.

To assess the effect of IR-induced DNA damage in the presence or absence of CHK2 knockdown, we performed comet assays to measure DNA breaks (*Collins, 2004*), and immunofluorescence staining of γH2AX, a marker for DNA double-strand breaks (*Vignard et al., 2013*; *Bonner et al., 2008*). When we examined DNA damage using the comet assay, we observed statistically significant less DNA breaks with CHK2 knockdown at 24 hr in LNCaP cells (*Figure 5B*; *Figure 5—figure supplement 1*). The comet assay shows less % DNA in tail, shorter DNA tail length, and lower tail moment

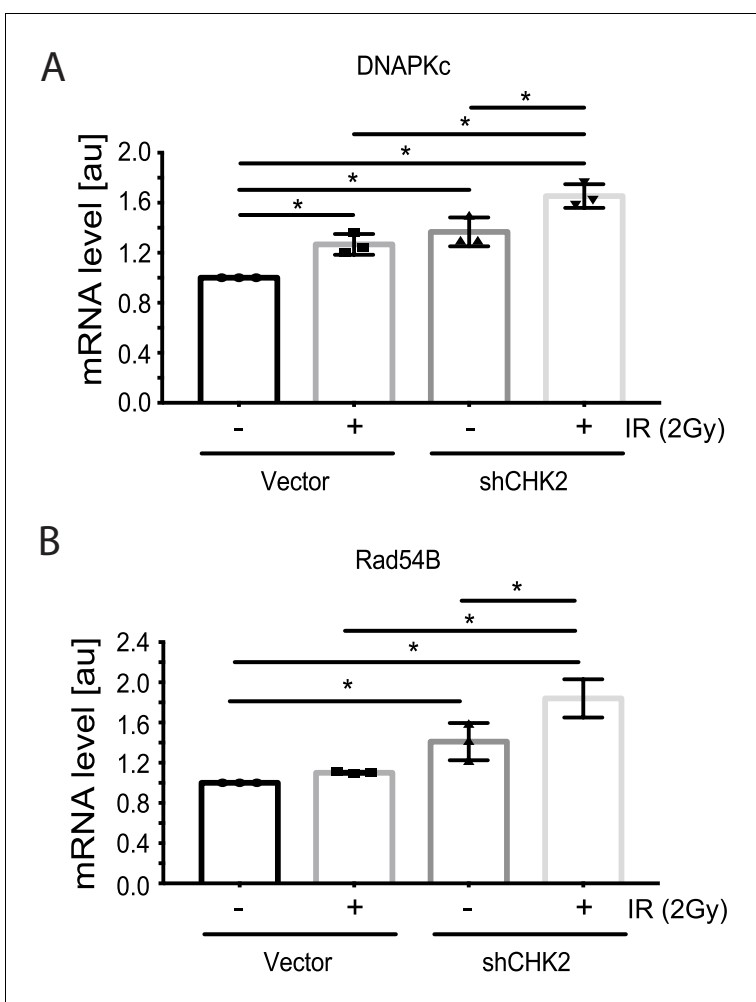

**Figure 4.** CHK2 knockdown increases the transcription of DDR genes in the presence and absence of radiation. Transcript levels of DDR genes in LNCaP cells transduced with CHK2 shRNAs and pLKO vector control and grown in CSS supplemented with 1 nM DHT were measured by qPCR. 48 hr following transduction, cells were exposed to 2Gy ionizing radiation and RNA was isolated 6 hr later. SQ means were determined using standard curves, normalized to the reference gene PSMB6, with the untreated pLKO condition set to 1x. Values were averaged across biological replicates +/- standard error of the mean, n = 3. Shown are the histograms for (**A**) DNAPKc and (**B**) Rad54B in LNCaP cells. Statistical analysis was performed using one-way ANOVA and Tukey's test. *=p < 0.02.

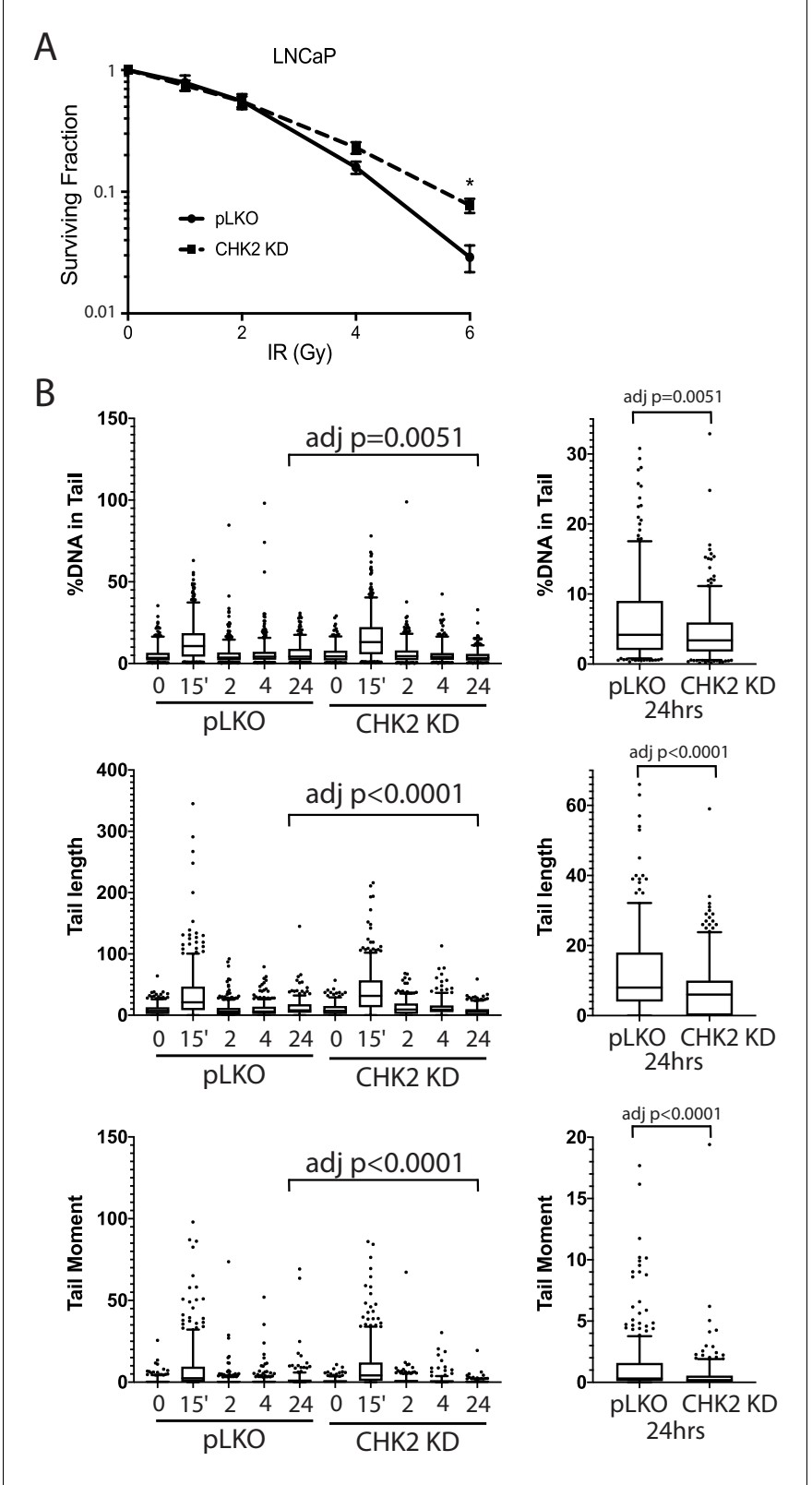

**Figure 5.** CHK2-depleted cells show increased survival and reduced DNA damage following radiation. (**A**) Knockdown of CHK2 desensitizes cells to IR. LNCaP cells were transduced with lentiviral particles expressing pLKO or CHK2 shRNAs for 48 hr, treated with 0-6Gy IR, and seeded at the appropriate cell number for colony survival assays. Results were normalized to untreated pLKO control and fitted to a standard linear quadratic model. Error

*Figure 5 continued on next page*

*Figure 5 continued*

bars, SEM. Statistical analysis was performed using the Student's t-test, n = 4–8, p<0.01. (**B**) LNCaP cells were transduced with lentiviral particles expressing pLKO or CHK2 shRNAs for 48 hr, irradiated at the indicated doses at the specified times and processed for comet assays. Shown is the %DNA in the comet tail, comet tail length, and tail moment (% DNA x tail length), n = 3 (209–592 cells measured per condition), 5–95% confidence intervals. Right most plots are of pLKO and CHK2 KD at 24 hr to illustrate the statistical difference marked on the graphs. Statistical differences were tested for using Kruskal-Wallis test followed by Dunn's multiple comparisons test. The online version of this article includes the following figure supplement(s) for figure 5:

**Figure supplement 1.** LNCaP cells were transduced with lentiviral particles expressing pLKO or CHK2 shRNAs for 48 hr, irradiated at the indicated doses at the specified times and processed for comet assays.

with CHK2 knockdown suggesting more efficient DNA repair when CHK2 is compromised. In order to quantify DSB repair activity in an unbiased manner, we developed an approach that utilizes automated quantitation of γH2AX and 53BP1 foci detected with immunofluorescence that enables us to rapidly determine the signal intensity (*Figure 6—figure supplement 1*). Rv1 cells were transduced with vector or CHK2 shRNA 48 hr before IR. Radiation induced more γH2AX foci at 15 min when CHK2 was knocked down (*Figure 6A and B*). However, less foci are found in CHK2 knock-down cells at 2 hr and by 4 hr through 24 hr an equal number of foci are present in control and CHK2 knock-down cells. H2AX is phosphorylated in response to inputs in addition to IR induced DNA double strand breaks, including RNA polymerase II dependent transcription (*Singh et al., 2015*; *Bunch et al., 2015*; *Ji et al., 2019*). Therefore, we also measured 53BP1 foci to assess DSB repair and found no significant differences when CHK2 was knocked down (*Figure 6C*). The apparent discrepancy in the 53BP1 and γH2AX foci may be explained by an increase in transcription when CHK2 is knocked down, as shown in *Figure 4* and in our previous study (*Ta et al., 2015*).

Since supraphysiological doses of androgen have been associated with transcription dependent double strand breaks (*Haffner et al., 2010*; *Chatterjee, 2019*), we also tested if CHK2 knockdown would augment supraphysiological androgen dependent γH2AX foci. LNCaP and Rv1 cells were transduced with vector or CHK2 shRNA 48 hr before treatment with a range of the synthetic androgen R1881 up to 100 nM for 6 hr (*Figure 6—figure supplement 2*). We saw modest hormone induced γH2AX foci and no change with CHK2 knockdown. The supraphysiologic dose of androgen likely maximally activates AR dependent transcription negating the increase in AR transcriptional activity typically observed with CHK2 knockdown.

## Discussion

PCa is the most frequently diagnosed cancer and the second leading cause of cancer death among American men, with approximately 88 men dying from PCa every day (pcf.org). While androgen deprivation therapy (ADT) is effective initially, most patients will relapse and develop incurable CRPC. Recently, there has been an emphasis on understanding the link between the DDR and AR, since radiation is a standard of care for locally advanced PCa where the AR is a major driver, and PARP inhibitors may be efficacious in CRPC patients with mutations in DDR genes (*Polkinghorn et al., 2013*; *Goodwin et al., 2013*; *Ta et al., 2015*; *Mateo et al., 2015*; *Li et al., 2014*; *Yin et al., 2017a*).

Our study identifies AR as a direct interacting protein with CHK2 in PCa cells. Several studies elucidated the role of DDR protein-AR interactions in modulating AR transcriptional activity. PARP-1 was recruited to AR binding sites, enhancing AR occupancy and transcriptional function (*Schiewer et al., 2012*). Tandem mass spectroscopy analysis identified Ku70 and Ku80 as direct AR-interacting proteins that positively regulate AR transactivation (*Mayeur et al., 2005*). Furthermore, BRCA1 physically interacted with the DNA-binding domain (DBD) of AR to enhance AR transactivation and induce androgen-mediated cell death through p21 expression (*Yeh et al., 2000*). In contrast, the association of the LBD of AR with hRad9, a crucial member of the checkpoint Rad family, suppressed AR transactivation by preventing the androgen-induced interaction between the n-terminus and c-terminus of AR (*Wang et al., 2004*). Other groups reported non-genomic effects as a result of DDR protein-AR interactions. Mediator of DNA damage checkpoint protein 1 (MDC1), an essential player in the Intra-S phase and G2/M checkpoints, physically associated with FL-AR and

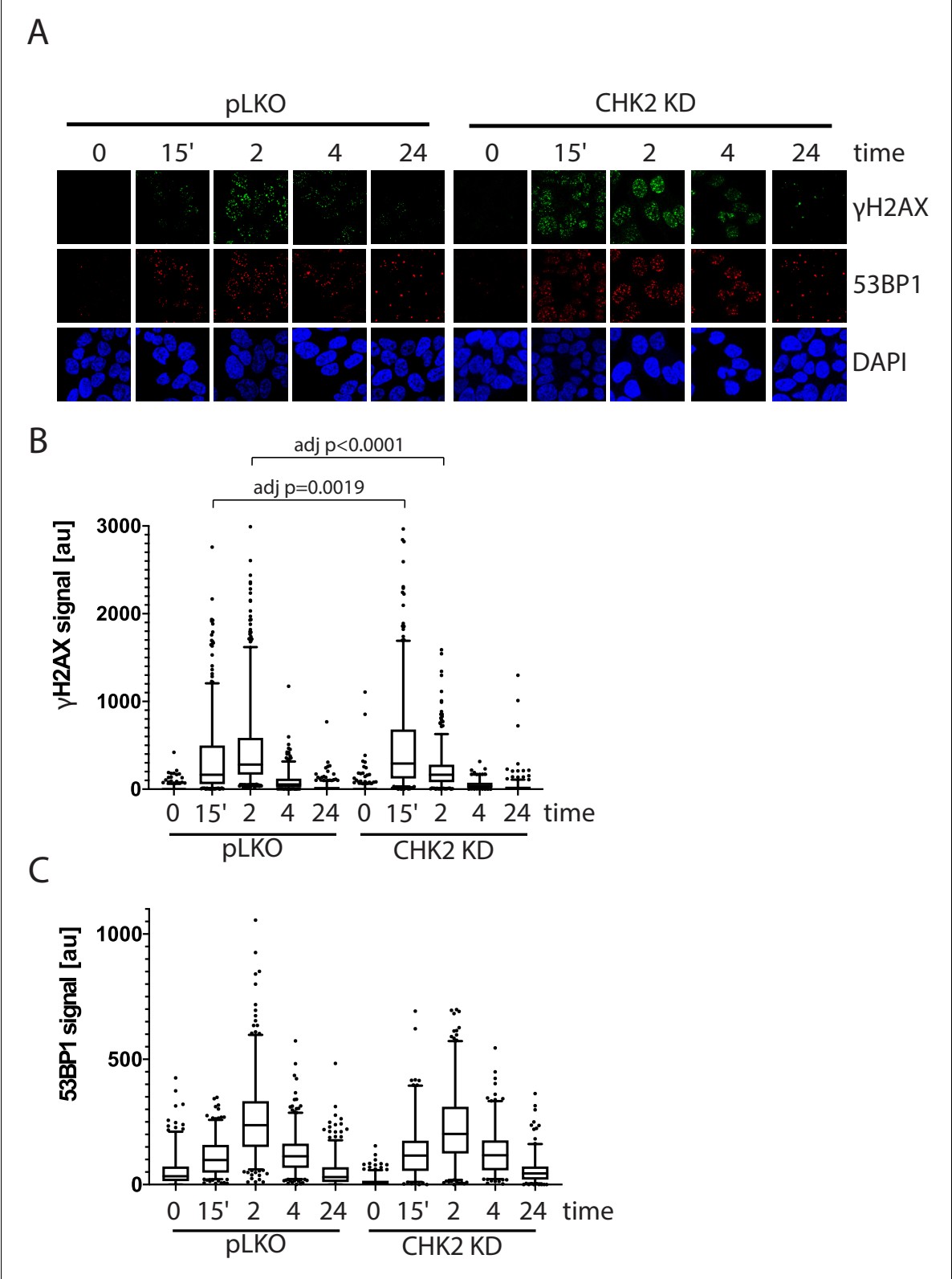

**Figure 6.** CHK2-depleted cells show altered γH2AX foci kinetics. (**A**) Representative images of γH2AX, CHK2, and AR immunostaining quantified in (**B** and **C**). (**B**) Rv1 cells were transduced with lentiviral particles expressing empty vector or CHK2 shRNAs on fibronectin-coated coverslips in the appropriate growth media. Cells were irradiated with 6Gy after 48 hr. Coverslips were processed for IF at 0, 15 min, 2, 4, and 24 hr following IR. Plotted is the (**B**) γH2AX signal and (**C**) 53BP1 signal, which equals the mean grey value intensity x number of foci per nucleus, n = 3 (378–565 nuclei measured

*Figure 6 continued on next page*

*Figure 6 continued*

per condition), 5–95% confidence intervals. Statistical differences were tested for using Kruskal-Wallis test followed by Dunn's multiple comparisons test.

The online version of this article includes the following figure supplement(s) for figure 6:

**Figure supplement 1.** Automated quantitation of foci.

**Figure supplement 2.** CHK2 Knockdown does not alter hormone induced γH2AX foci.

AR[V7] to negatively regulate PCa cell growth and migration (*Wang et al., 2015*). Yin and colleagues, on the other hand, showed that increased clonogenic survival following IR was a consequence of DNA-PKc directly complexing with both FL-AR and AR[V5-7], with radiation increasing these interactions and enzalutamide blocking the association with FL-AR but not AR[V5-7] (*Yin et al., 2017b*). These data support a model where AR is integrated in the DDR, interfacing at multiple points.

We show that the direct association of CHK2 and AR by Far western requires phosphorylation of AR on S308. Since proteins containing FHA domains bind phosphoproteins (*Li et al., 2002*), we hypothesize that AR interacts with CHK2 through the CHK2 FHA domain. In support of this, the Zhao lab determined that AR physically associated with the FHA domain of another critical DDR member, MDC1 (*Wang et al., 2015*). Expression of truncation mutants of different MDC1 domains in LNCaP cells led to the discovery that AR only co-immunoprecipitated with MDC1 mutants containing the FHA domain in the absence and presence of dihydrotestosterone. Their results indicated that the FHA domain of MDC1 mediated the interaction with AR.

Here we report that AR phosphorylation on S81 and S308 is required for the IR induction of AR – CHK2 binding by co-IP. Interestingly neither of these phosphorylation sites were altered by IR. AR S81 and S308 can both be phosphorylated by CDK1, which is downstream of canonical CHK2 signaling (*Chen et al., 2012*; *Koryakina et al., 2015*), and was the motivation for examining these sites in response to IR. The prediction is that IR would lead to a decrease in S81 and S308 phosphorylation. However, our previous studies demonstrated that S81 is predominantly phosphorylated by CDK9, and thus is more indicative of AR transcriptional activity (*Gordon et al., 2010*). We also found that S308 phosphorylation was restricted to late G2 and M phase of the cell cycle (*Koryakina et al., 2015*). CDK9 phosphorylation of S81 and the restriction of S308 phosphorylation to G2/M likely accounts for the lack of significant change in these phosphorylation sites in response to IR. The disconnect between these sites being required for the IR induced increase in AR – CHK2 association but not being regulated by IR suggests that the hormone induced activation state of the AR is a critical determinant in the IR induced increase in AR – CHK2 association. We observed differences when testing the AR – CHK2 association by Far western and co-IP. The direct protein-protein interaction is only impacted by loss of S308 phosphorylation, however, basal co-IP of AR and CHK2 is not affected by AR phosphorylation. Only the IR induced AR – CHK2 association is impacted by loss of AR phosphorylation on S81 and S308. This suggests that additional proteins present in cells may assist and participate in the AR – CHK2 interaction.

We found that CHK2 variants with diminished kinase activity impaired the IR-induced increase in AR – CHK2 interaction but did not completely block the interaction. As with AR phosphorylation, there were differences when the CHK2 mutant association with AR was examined by Far western or co-IP suggesting a more complicated dynamic of AR – CHK2 association in cells. The PCa associated CHK2 mutants also exhibited a reduced inhibition of cell growth. CHK2-depleted cells re-expressing CHK2 variants exhibited an approximate 2–3-fold reduction in growth inhibition in response to hormone when compared to cells re-expressing wtCHK2. Moreover, the fold change in growth suppression between wtCHK2 and CHK2 variants was greater in androgen-dependent LNCaP cells than in castration-resistant C4-2 and Rv1 cells, suggesting that in hormone sensitive PCa CHK2 variants may play a larger role in regulating growth. This raises the possibility that the CHK2 mutants found in PCa with diminished AR binding are selected for, decreasing CHK2 suppression of AR activity and PCa cell growth. Berge and colleagues discovered numerous CHK2 splice variants in breast cancer tissue, where all variants were co-expressed with wtCHK2 (*Berge et al., 2010*). Furthermore, several of these variants reduced kinase activity when simultaneously expressed with wtCHK2 and displayed a dominant negative effect on wtCHK2. The impact of the CHK2 variants found in PCa on wtCHK2 function has not yet been fully explored.

In our experiments examining AR – CHK2 binding, ERK was used as a positive control for a protein that interacts with CHK2 (*Dai et al., 2011*). Interestingly, we observed a significant increase in CHK2–ERK association with IR. This is consistent with IR increasing CHK2 T68 phosphorylation, which is required for the CHK2–ERK interaction. These data point to a potentially larger role for CHK2 beyond canonical DDR and cell cycle checkpoint signaling; consistent with this notion CHK2 has been implicated in diverse cellular processes (*Zannini et al., 2014*; *Choi et al., 2014*; *Choudhuri et al., 2007*). MEK inhibition is effective in lung tumors with ATM mutations where CHK2 is inactive (*Smida et al., 2016*) providing further support that CHK2 negatively regulates ERK. Negative regulation of both the AR and ERK by CHK2 suggests the hypothesis that CHK2 may serve as a general negative regulator of mitogenic signals in response to IR.

Multiple studies indicate that the AR is a critical regulator of genes in the DDR (*Polkinghorn et al., 2013*; *Goodwin et al., 2013*). Reports by others demonstrate that androgen and IR increased DNAPK, XRCC2, and XRCC3 (*Goodwin et al., 2013*). This concept was supported by more global analysis of transcripts demonstrating androgen regulation of DDR genes (*Polkinghorn et al., 2013*). Another global analysis identified a different set of DDR genes are regulated by AR (*Jividen et al., 2018*). Our test by RT-qPCR of ten different DDR genes found only a modest induction by IR or androgen treatment in two DDR genes, DNAPK and RAD54. The discrepancy in the DDR gene subsets regulated by the AR in these studies and ours may be system dependent. However, it also indicates that the dogma of AR regulating DDR genes needs further examination. Our data indicate that CHK2 knockdown increases DNAPK and RAD54 transcript levels is consistent with our earlier observation that CHK2 knockdown led to the increase in the transcripts of canonical AR target genes (*Ta et al., 2015*). This leads to the hypothesis that CHK2 binding to the AR suppresses AR transcription of DDR genes enabling cells to turn off the DDR following DNA repair.

We observed radiation resistance in CHK2 knockdown cells. We also observed an increase in γH2AX signal when CHK2 was knocked down, and a decrease in DNA breaks as measured by the comet assay under similar conditions. However, when 53BP1 signal was measured to assess DSB repair we found no significant differences when CHK2 was knocked down. These conflicting results of γH2AX and 53BP1 signal may be explained by phosphorylation of H2AX in response to transcription induced DNA breaks (*Singh et al., 2015*; *Bunch et al., 2015*; *Ji et al., 2019*). These incongruous results may also be due to competing effects of CHK2 as both a regulator of the cell cycle and apoptosis (*Zannini et al., 2014*).

The data reported herein along with our previous work (*Ta et al., 2015*) indicate that CHK2 acts as a tumor suppressor in PCa, either through loss of expression or mutation. This raises the concern that CHK2 antagonists in clinical development may paradoxically lead to enhanced PCa growth and resistance to IR. However, it is important to note that we have predominately used an RNAi/overexpression approach. Our RNAi approach is more similar to the CHK2 variants in PCa that have reduced kinase activity. It is important to consider that pharmacologic inhibition is different than inhibition by RNAi (*Jensen et al., 2016*). A pharmacologic approach that provides a sudden and complete inhibition of CHK2 kinase activity may impact PCa differently than our RNAi approach, especially when combined with IR or AR antagonists. However, our work and that in the literature also suggests that approaches downstream of CHK2 may be more straightforward than targeting CHK2.

In this study, we presented data that provides mechanistic insight into our observation that CHK2 negatively regulates PCa growth. We demonstrated that AR directly bound CHK2, and that IR elevated the AR – CHK2 interaction. Not only did these AR – CHK2 protein complexes require AR phosphorylation, but CHK2 kinase activity was also necessary. Furthermore, these CHK2 mutants found in PCa exhibited a diminished effect on restricting prostate cancer cell growth. We observed that knockdown of CHK2 led to an increase in PCa cell survival and resolution of DNA DSBs in response to IR. This suggests that the deregulation of CHK2 in PCa can confer resistance to radiation. In a previous study, we showed that CHK2 knockdown hypersensitized PCa cells to castrate levels of androgen and increased AR transcriptional activity on both androgen-activated and androgen-repressed genes (*Ta et al., 2015*). As part of a feedback loop, AR transcriptionally represses CHK2 levels. Thus, these data along with our previously published results suggest the following model (*Figure 7*). Wild-type CHK2 maintains homeostasis through mediating the balance between DNA repair and cell death, antagonizes the AR through direct binding and inhibition of transcription of AR target genes

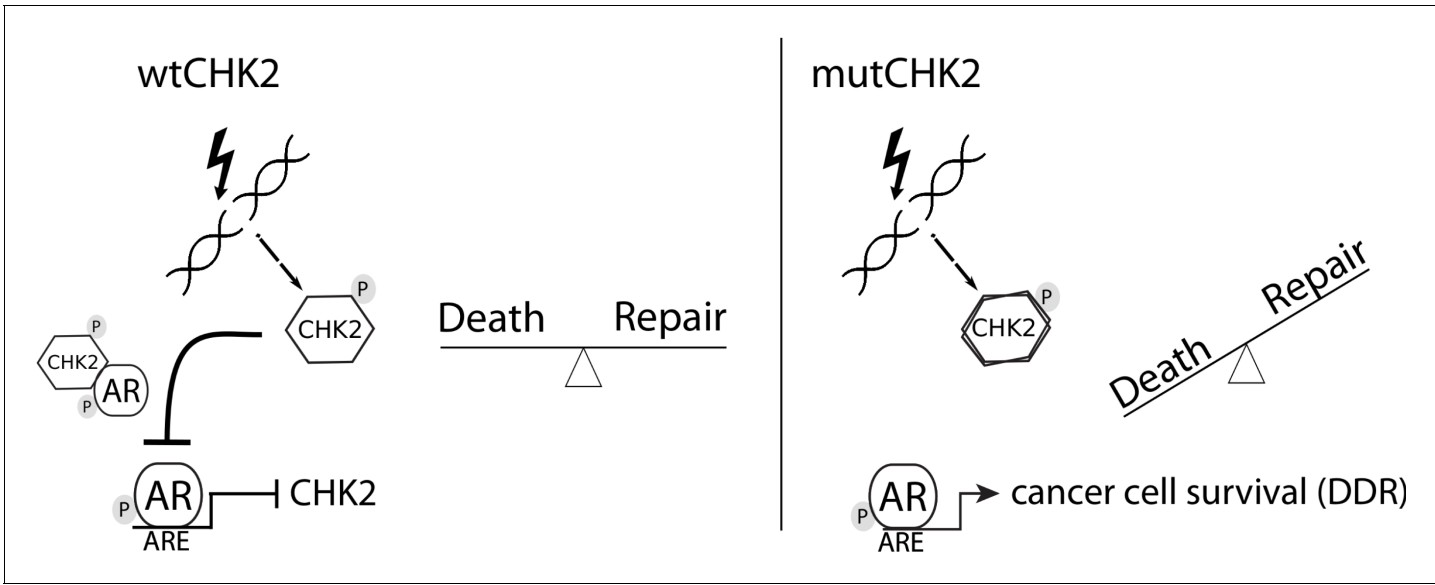

**Figure 7.** Model of CHK2-AR. We hypothesize the following model. In response to IR, CHK2 activation antagonizes AR through direct binding and inhibition of transcription of AR targets. CHK2 mutation or loss of expression that occurs in PCa leads to sustained AR transcriptional activity, an increase in DDR gene transcripts, and survival in response to DNA damage.

while AR transcriptionally represses CHK2. This feedback loop enables AR repression of CHK2 to turn off the DDR, while CHK2 negatively regulates AR to dampen growth during repair of damaged DNA. CHK2 mutation or loss of expression in PCa leads to increased AR transcriptional activity, possibly of DDR genes, and survival in response to DNA damage, all leading to a more aggressive cancer. Collectively, the work provides a foundation for the continued study of AR – CHK2 interactions and functional consequences to benefit PCa therapies.

## Materials and methods

### Cell culture

LNCaP (RRID:CVCL_0395) and C4-2 (RRID:CVCL_4782) cells (a gift from Dr. L. W. K. Chung) were grown in DMEM:F12 (Invitrogen) with 5% non-heat inactivated serum (Gemini) and 1% Insulin-Transferrin-Selenium-Ethanolamine (ITS) (ThermoFisher). CWR22Rv1 (Rv1) (RRID:CVCL_1045) (gift from Drs. Steven Balk) and 293T (RRID:CVCL_0063) cells (gift from Dr. Tim Bender) were grown in DMEM (Invitrogen) with 10% heat-inactivated serum. For growth experiments, phenol-red free DMEM:F12 media with 5% Charcoal-Stripped Serum (CSS) (Sigma) was used. Commercial DNA fingerprinting kits (DDC Medical) verified cell lines. The following STR markers were tested: CSF1PO, TPOX, TH01, Amelogenin, vWA, D16S539, D7S820, D13S317 and D5S818. Allelic score data revealed a pattern related to the scores reported by the ATCC, and consistent with their presumptive identity. Cell lines are tested monthly for mycoplasm or three passages following reconstitution from frozen stocks.

### Reagents

Transfection: Fugene 6 (Promega); TransIT-2020 (Mirus Bio). Inhibitors: Enzalutamide (Selleck Chemicals), BML-277 (Santa Cruz Biotech). Antibodies: CHK2 (2G1D5), pCHK2 T68, ERK1/2 (137F5), Actin, FLAG-Tag, V5-Tag, HA-Tag, γH2AX (Cell Signaling); AR, pAR S308 (in-house); pAR S81 (Millipore); Cy3-labeled donkey anti-rabbit (Jackson ImmunoResearch). Western blotting performed as previously described (*Ta et al., 2015*).

### Far western blot

To measure direct protein interactions, the protocol was adapted from *Prickett et al., 2008* and *Wu et al., 2007*. 293 T cells were transfected with FLAG-wtAR, FLAG-S81A, S81D, S308A, S308D,

HA-wtAR, FLAG-ERK2, V5-wtCHK2, FLAG-CHK2, FLAG-CHK2 mutants K373E or T387N, or empty vector control. Whole cell extracts were made using Triton-X lysis buffer, sonicated, and immuno-purified using anti-FLAG, anti-HA, or anti-V5 beads (Sigma) for 2 hr at 4°C. Protein bound to beads was washed three times with Triton-X lysis buffer, eluted with 35 µl 2X sample buffer, and boiled for 5 min. Proteins were resolved by SDS-PAGE and transferred to PVDF membrane. Proteins on the membrane were denatured and renatured in buffers with varying guanidine–HCl concentrations. Membranes were blocked in 3% blocking buffer (3% bovine serum albumin in Tris-buffered saline/ Tween 20) for 1 hr. Probes were diluted in 3% blocking buffer and incubated overnight at 4°C. Mem-branes were washed three times with PBS for 5 min followed by fixation using 0.5% paraformalde-hyde for 30 min at room temperature. Membranes were then rinsed quickly twice with PBS and quenched using 2% glycine in PBS for 10 min at room temperature. The membrane was blotted for FLAG, HA, V5, AR, CHK2, or ERK1/2 and analyzed using the LI-COR Odyssey system and software.

## Immunoprecipitation

CHK2 or AR protein was immunoprecipitated from 1 mg cell lysate from LNCaP cells transiently transfected with FLAG-wtAR/FLAG-S81A/FLAG-S308A or FLAG-wtCHK2/Flag-K373E/Flag-T387N plus HA-wtAR for 48 hr; treated with radiation. Immunoprecipitations (IP) were performed with either agarose or magnetic beads, proteins were separated by 7.5% SDS-PAGE; and immunoblotted with AR, pAR S81, pAR S308, CHK2, pCHK2 T68, HA, or ERK1/2 antibodies. Non-cropped versions of the Far western and IP western blots are shown in *Figure 3—figure supplement 1*.

## CyQuant growth assays

Assay was performed as previously described (*Ta et al., 2015*). Briefly, shCHK2-209, shCHK2-588, wtCHK2, K373E, T387N or Vector control virus was added to fibronectin-coated (1 µg/ml) 96well plates. Constructs of CHK2 wild-type and variants were verified by sequencing. Cells were plated in phenol-red free DMEM:F12 or DMEM media with 5% CSS in the presence or absence of 0.05nM R1881. CyQuant reagent was added on day 7 according to the manufacturer's protocol (Thermo-Fisher). Quantification was performed using a BioTek Synergy 2 plate reader.

## qPCR

RNA isolation and quantitative real-time PCR (qPCR) were performed as previously described (*Gordon et al., 2010*; *Whitworth et al., 2012*). RNA concentrations were determined using a Nano-Drop 2000 UV-Vis Spectrophotometer (Thermo Scientific). Primer sequences and annealing tempera-ture: DNAPKc FW (ATGAGTACAAGCCCTGAG); DNAPKc RV (ATATCAGAGCGTGAGAGC) (Tm = 60 deg). RAD54B FW (ATAACAGAGATAATTGCAGTGG); RAD54B RV (GATCTAATG TTGCCAGTGTAG) (Tm = 60 deg). PSMB6 FW (CAAACTGCACGGCCATGATA); PSMB6 RV (GAGGCATTCACTCCAGACTGG) (Tm = 60 deg). The relative standard curve method was used to determine transcriptional fold changes as we have done previously (*Wong and Medrano, 2005*; *Bustin, 2000*; *Gordon et al., 2017*; *Whitworth et al., 2012*). RNA starting quantities (SQ) were determined using a standard curve. The SQ mean of DNAPKc or RAD54B was then normalized to the reference gene PSMB6.

## Clonogenic survival assay

LNCaP cells were transduced with lentiviral particles expressing vector or CHK2 shRNAs and treated with 0-6Gy of radiation 72 hr after transduction. Cells were trypsinized, counted, and appropriate numbers were plated in triplicate with the appropriate growth media for colony formation assays (100 cells/0Gy, 200 cells/2Gy, 1000 cells/4Gy, and 6000 cells/6Gy). After 10–14 days, colonies con-sisting of 50–70 cells were counted using crystal violet. All counts were made by a single observer in a blinded manner using a colony counter pen to maximize consistency and minimize bias. Plotted is the surviving fraction (number of colonies counted/(number of cells seeded x PE) where PE = plating efficiency=number of colonies counted/number of cells seeded) following radiation.

## Comet assay

All Comet Assay steps were performed in the dark. Cells were washed twice with PBS, scraped and suspended in PBS. Cells were combined with molten LMAgarose (Trevigen, 4250-050-02) and placed

into comet suitable slides. Samples were left at 4°C for 15 min in order to create a flat surface. Slides were immersed in lysis solution (Trevigen, 4250-050-01) for 40 min at 4°C and then in alkaline solution for 30 min at room temperature. Slides were electrophoresed in 200 mM NaOH, 1 mM EDTA in water; pH >13 at 21Volts (300mA) for 30 min at 4°C. After the electrophoresis, slides were gently drained, washed twice in dH2O for 10 min and immersed in 70% ethanol for 5 min. Samples were left to dry overnight at RT. Samples were stained with SYBR Green at 4°C for 5 min and left to dry completely overnight. For quantification, images were acquired using a fluorescence microscope (Olympus BX51, High-mag) equipped with a 20×, 0.5 NA objective and a camera (DP70). Images were acquired with DPController software. Images were analyzed by ImageJ software and graphs generated using Prism (GraphPad Software). All imaging was performed at ~24°C.

## Immunofluorescence (IF)

LNCaP cells were transduced with lentivirus expressing vector or CHK2 shRNAs on 1 µg/ml fibronectin-coated coverslips and treated with radiation 48 hr after transduction. Cells were allowed to recover from IR exposure up to 24 hr. Coverslips were washed 3X with PBS, permeabilized with 0.2% Triton-X for 5mins, washed 3x quick and 3 × 5 min with PBS-Tween 20 at 0.05% (PBS-T), blocked with 2% FBS/BSA/donkey serum in PBS-T for 2 hr at room temperature, and incubated with γH2AX or 53BP1 antibodies overnight at 4°C. Coverslips were then washed by rocking for 3x quick and 3 × 10 min in PBS-T and incubated with donkey anti-rabbit or anti-mouse secondary antibody (Invitrogen) for 1 hr at room temperature. Coverslips were then washed by rocking for 3x quick and 3 × 10 min in PBS-T and mounted with Vectashield containing DAPI (ThermoFisher). Images were acquired with a LSM 880 confocal microscope (Carl Zeiss) and signals were measured using ImageJ software.

## Scientific rigor

Each experiment was performed independently a minimum of three times and each experiment had technical replicates for measuring the endpoint. An independent experiment is defined as an experiment performed on a different day with a different cell line passage number. The number of independent experiments is reported in each figure legend. All data are shown, no outliers were removed. Statistical analysis was performed using GraphPad Prism 8.2.1 and the test used is reported in each figure legend.

# Acknowledgements

We thank the members of the laboratories of Drs. Gioeli, Jameson, Bouton, Dudley, Kashatus, Park, Rutkowski, Smith, and Zong for helpful discussions. We would also like to acknowledge exceptional assistance from UVA's Advanced Microscopy Facility.

# Additional information

### Funding

| Funder | Grant reference number | Author |
| --- | --- | --- |
| NIH Office of the Director | R01 CA178338 | Daniel Gioeli |
| Paul Mellon Urologic Cancer Institute | | Daniel Gioeli |
| University of Virginia Cancer Center Patient and Friends | | Daniel Gioeli |

The funders had no role in study design, data collection and interpretation, or the decision to submit the work for publication.

### Author contributions

Huy Q Ta, Data curation, Formal analysis, Visualization, Methodology, Writing-original draft; Natalia Dworak, Data curation, Formal analysis, Visualization; Melissa L Ivey, Data curation, Formal analysis;

Devin G Roller, Data curation, Formal analysis, Writing - review and editing; Daniel Gioeli, Conceptualization, Resources, Formal analysis, Supervision, Funding acquisition, Visualization, Methodology, Writing - original draft, Project administration, Writing - review and editing

**Author ORCIDs**
Daniel Gioeli (ID) https://orcid.org/0000-0003-2284-8152

**Decision letter and Author response**
Decision letter https://doi.org/10.7554/eLife.51378.sa1
Author response https://doi.org/10.7554/eLife.51378.sa2

## Additional files

### Supplementary files
• Transparent reporting form

### Data availability
Source data are available via the Open Science Framework (https://osf.io/2bx5q/).

The following dataset was generated:

| Author(s) | Year | Dataset title | Dataset URL | Database and Identifier |
|---|---|---|---|---|
| Gioeli D | 2020 | AR_CHK2_Gioeli | https://osf.io/2bx5q/ | Open Science Framework, 2bx5q |

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
