## [Decision Letter]

**Acceptance summary:**

The manuscript presents novel and compelling data on the interaction between CHK2 and Androgen Receptor in prostate cancer cells, by which CHK2 suppresses prostate cancer cell growth. These data explain how CDK2 mutations in prostate cancer lead to increased androgen receptor activity and tumor cell survival in response to DNA damage induction.

**Decision letter after peer review:**

Thank you for submitting your article "Checkpoint kinase 2 regulates prostate cancer cell growth through physical interactions with the androgen receptor" for consideration by *eLife*. Your article has been reviewed by three peer reviewers, including Wilbert Zwart as the Reviewing Editor and Reviewer #1, and the evaluation has been overseen by Jonathan Cooper as the Senior Editor. The following individual involved in review of your submission has agreed to reveal their identity: Kavita Shah (Reviewer #3).

The reviewers have discussed the reviews with one another and the Reviewing Editor has drafted this decision to help you prepare a revised submission.

In this study, Gioeli and colleagues build on their 2015 Cancer Research paper showing that AR transcriptionally represses CHK2 and its negative effects on PCa growth and androgen sensitivity, by investigating mechanisms underlying this CHK2/AR cross-talk in the specific context of radiation-induced signaling and DNA damage response. They propose that their findings indicate that wild-type CHK2 can sequester AR and prevent its transcriptional enhancement of DNA repair genes, which in turn confers PCa radioresistance and survival under DNA damage.

Overall this is an interesting and conceptually novel study. The authors convincingly show that AR and CHK2 interact physically, that IR transiently stabilizes this interaction, and that AR activity is required for this interaction. However, a number of issues need to be addressed in a revised manuscript:

Essential revisions:

1) Concerns were raised on the experimental design and interpretation of data in Figure 8 and Supplementary Figure 2. Assuming the model in Figure 9 is correct, then under IR treatment, CHK2 depletion should significantly decrease unrepaired DNA breaks by increasing the frequency/efficacy of DNA repair through AR activity. This should manifest, over time, in fewer unresolved γH2AX foci and *lower* %tail DNA/tail moments relative to the pLKO controls, indicative of radioresistance. Although the AR/CHK2 interaction peaks at 1 hour, this is not the optimal time frame in which to evaluate the levels of unrepaired DNA breaks which would compromise PCa growth and survival by triggering tumor-inhibitory responses to radiation. γH2AX typically appear as punctate foci 6-12 hours following DNA damaging stresses. The diffuse staining in Figure 8B indicates the timepoints of detection are prior to formation of DDR complexes at DSB sites or potentially technical issues (typically optimal detection of this epitope requires stringent washing and antibody incubation in a detergent-based buffer to minimize the degree of background seen here). Either issue calls the accuracy of foci quantitation into question.

2) The alkaline comet assay detects not just DNA double strand breaks but also DNA nicks and abasic sites (through initiation of base excision repair) which will abound following IR treatment. To assess whether these have been repaired properly or not, the assay should be repeated at later timepoints (12-18 hours) when the changes in %tail DNA and tail moments will better represent efficiency and extent of repair. The authors may find a far more striking difference between their control and CHK2-depleted cells, if they repeat the γH2Ax experiment at 12-18 hours following IR when the punctate foci (indicative of repair efficiency or lack thereof) will become more apparent. The difference in comet tail DNA metrics will also most clearly reflect unrepaired genotoxic DNA intermediates at a longer time point. The data, as they stand, do not support their model. Similarly, the relatively small changes in DNA PKc and RAD54 transcripts in Figure 7 do not convince of their assertion that CHK2 depletion increases PCa survival through AR-mediated transcription of DNA repair genes. Such transcript-level changes alone in these factors cannot be extrapolated to their functional effects on DNA repair. The increase in clonogenic survival 14 days post-IR in CHK2-depleted cells is fairly modest and does not necessarily support that CHK2 inhibits radioresistance (Figure 8A). As the authors show in Figure 6, LNCaP cells are generally more growth inhibited by wt CHK2 so the differences in Figure 8A may not reflect radioresistance but just a general growth advantage upon CHK2 loss.

3) It is unclear what PCa cell growth actually refers to in this study – is the reduction in cell numbers due to a proliferation arrest or cell death? The known CHK2 interactomes could affect either (or both) pathway. Some textual and/or experimental elaboration of this point is warranted, considering this is a central premise of the study.

4) The findings from this study taken in context with the previous 2015 study would suggest that the CHK2/AR interaction is mutually inhibitory, as the earlier study suggested that CHK2 is transcriptionally repressed by AR. However, this mutual repression and the notion of negative feedback regulation from CHK2 to AR is not discussed here, for instance in the model in Figure 9. This aspect should be addressed, at least textually, to provide a complete picture of this crosstalk in PCa.

5) "Kinase-impaired CHK2 variants, including the K373E variant associated with 4.2% of PCa, *blocked* IR-induced CHK2-AR interactions". This is a strong statement considering the blot in Figure 3B doesn't appear representative of the quantitation shown – it seems the variants reduce binding but certainly don't seem to block it completely.

6) It seems IR increases the CHK2/AR interaction by increasing CHK2 phosphorylation and therefore its activity, which is an interesting finding. However, both the experimental design and Figure 4 (which describes this effect) are hard to understand. For example, the blots on the left vs. the respective quantitation on the right are confusing. The graphs look different (- or + radiation of the V5-CHK2 cells) but the blots look very similar. It would help if Figure 4 included a schematic of the experimental design.

7) Statistical analyses and quantifications should be provided for all figure and supplementary panels, when appropriate (e.g. Figure 5A, Figure 8C, Supplementary Figure 2, Figure 6—figure supplement 1).

8) Did the authors check whether CHK2 directly phosphorylates AR, and whether this phosphorylation is required for their binding?

9) Although the authors show that CHK2 binding with AR peaks at 1 hour, it would be ideal if the others would have tested additional early time points (such as 30 min, 45 min, 1.5h) to exactly determine the peaking of signal. Are these data available?

10) As ARS81 is phosphorylated by CDK1, does CDK1 inhibition (using small molecule inhibitors) impact IR induced association of CHK2 and AR?

11) Please check whether the ERK1/2 control blots in Figure 6B (C42) vs. 6C (22Rv1) have been accidentally duplicated.

---

## [Author Response]

Essential revisions:1) Concerns were raised on the experimental design and interpretation of data in Figure 8 and Supplementary Figure 2. Assuming the model in Figure 9 is correct, then under IR treatment, CHK2 depletion should significantly decrease unrepaired DNA breaks by increasing the frequency/efficacy of DNA repair through AR activity. This should manifest, over time, in fewer unresolved γH2AX foci and lower %tail DNA/tail moments relative to the pLKO controls, indicative of radioresistance. Although the AR/CHK2 interaction peaks at 1 hour, this is not the optimal time frame in which to evaluate the levels of unrepaired DNA breaks which would compromise PCa growth and survival by triggering tumor-inhibitory responses to radiation. γH2AX typically appear as punctate foci 6-12 hours following DNA damaging stresses. The diffuse staining in Figure 8B indicates the timepoints of detection are prior to formation of DDR complexes at DSB sites or potentially technical issues (typically optimal detection of this epitope requires stringent washing and antibody incubation in a detergent-based buffer to minimize the degree of background seen here). Either issue calls the accuracy of foci quantitation into question.

We thank the reviewers for suggesting that we look at later times for DNA damage and DNA repair activity. When we examine DNA damage using the comet assay we do in fact observe statistically significantly less DNA breaks with CHK2 knockdown at 24hrs. The comet assay shows less % DNA in tail, shorter DNA tail length, and lower tail moment with CHK2 knockdown. Interestingly, we observe more γH2AX foci early and quicker resolution of the foci with CHK2 depletion, suggesting a more rapid resolution of the DNA damage when CHK2 is knocked down. These data are now included in the resubmitted manuscript.

With regards to the kinetics of DNA repair following irradiation, data in the literature demonstrate that DSB repair has short half-life of 10 to 60 minutes depending on the type of DNA repair utilized (PMIDs: 1677379, 7841046, 10728683, 12643795, 17088286). DSBs are also repaired faster in cancer cells (PMID: 3182330), DSB repair is fast when NHEJ is utilized (PMIDs: 21530420), and the CWR22Rv1 cells have a mutation in BRCA2 and the LNCaP lineage displays “BRCAness” (PMIDs: 28536297, 29465803, 30012171), which results in reliance on the more rapid NHEJ. Additionally, cells in G1 cannot use HR and will use NHEJ to repair DSBs and the majority of the PCa cells are in G1. Furthermore, NHEJ is utilized in G2 (PMID: 18155970). Thus, given the system that we are using, the data in the literature supports the importance of looking at early time points for DSBs and DNA repair foci.

In response to IR, γH2AX foci can be cleared in a little as 3 hours (PMIDs: 16546909, 29127285, 25786477), although as noted by the reviewers, foci can remain for much longer and importantly, the remaining low number of foci may reflect unresolved DSB repair that could compromise PCa cell growth/survival. This is reflected in our observation of a statistically significant reduction in DSBs in the CHK2 knockdown. We have also extended our time course to 24hrs and have now included data for both γH2AX and 53BP1 foci in the resubmitted manuscript.

We have rigorously optimized foci staining (no primary/no secondary controls) and do our antibody incubations in detergent containing buffer. We have added additional details to our Materials and methods for clarity. We have also developed an unbiased method for foci quantitation as described in Figure 6—figure supplement 1. We strongly believe the reported data are an accurate reflection of the γH2AX and 53BP1 foci in PCa cells in response to IR.

2) The alkaline comet assay detects not just DNA double strand breaks but also DNA nicks and abasic sites (through initiation of base excision repair) which will abound following IR treatment. To assess whether these have been repaired properly or not, the assay should be repeated at later timepoints (12-18 hours) when the changes in %tail DNA and tail moments will better represent efficiency and extent of repair. The authors may find a far more striking difference between their control and CHK2-depleted cells, if they repeat the γH2Ax experiment at 12-18 hours following IR when the punctate foci (indicative of repair efficiency or lack thereof) will become more apparent. The difference in comet tail DNA metrics will also most clearly reflect unrepaired genotoxic DNA intermediates at a longer time point. The data, as they stand, do not support their model. Similarly, the relatively small changes in DNA PKc and RAD54 transcripts in Figure 7 do not convince of their assertion that CHK2 depletion increases PCa survival through AR-mediated transcription of DNA repair genes. Such transcript-level changes alone in these factors cannot be extrapolated to their functional effects on DNA repair. The increase in clonogenic survival 14 days post-IR in CHK2-depleted cells is fairly modest and does not necessarily support that CHK2 inhibits radioresistance (Figure 8A). As the authors show in Figure 6, LNCaP cells are generally more growth inhibited by wt CHK2 so the differences in Figure 8A may not reflect radioresistance but just a general growth advantage upon CHK2 loss.

We are very appreciative of the suggestion to look at later time points for the comet assay. As mentioned above in our response to the first comment, when we look at 24hrs the comet assay shows less % DNA in tail, shorter DNA tail length, and lower tail moment with CHK2 knockdown. This data is now included in the manuscript.

The reviewers’ point that the changes in DNA PKc and RAD54 transcripts when CHK2 is knocked down may be insufficient to drive our observations of increased DNA repair and survival with CHK2 knockdown in PCa is valued. This has been a point of discussion in my laboratory since we first examined the level of DDR transcripts reported in PMIDs: 24027197 and 24027196 and only observed differences in DNA PKc and RAD54. Another study (PMID: 30305041) showed that a different set of DDR genes is regulated by AR. The discrepancy in the DDR gene subsets regulated by the AR in these studies may be system dependent, however, it also indicates that the dogma that AR regulates DDR gene expression needs further examination. We have added text in the Results and Discussion of the manuscript that we feel more completely addresses this issue.

3) It is unclear what PCa cell growth actually refers to in this study – is the reduction in cell numbers due to a proliferation arrest or cell death? The known CHK2 interactomes could affect either (or both) pathway. Some textual and/or experimental elaboration of this point is warranted, considering this is a central premise of the study.

For our earlier study when we first described that CHK2 functioned as a tumor suppressor gene and that CHK2 knockdown increased cell growth, we performed preliminary studies of the cell cycle distribution based on DNA content and apoptosis by western blotting of cleaved caspase and PARP. Normal growing PCa cells have low to not detectable levels of cleaved caspase and PARP, making it difficult to determine if there was a reduction in apoptosis with CHK2 knockdown. This would favor a model whereby CHK2 knockdown leads to increased proliferation. Preliminary cell cycle analysis by flow cytometry in LNCaP cells +/- CHK2 KD indicated an increase in S and G2/M, however, this would have to be explored further to be conclusive. We have not examined proliferation/death in the context of IR. Since DSBs induce apoptosis we expect the increased clonagenic survival with CHK2 knockdown is due to the increase in DSB repair (supported by the comet assay, see response to concerns #1 and 2) and corresponding reduction in DSB-induced apoptosis. We have added text to the manuscript addressing this point.

4) The findings from this study taken in context with the previous 2015 study would suggest that the CHK2/AR interaction is mutually inhibitory, as the earlier study suggested that CHK2 is transcriptionally repressed by AR. However, this mutual repression and the notion of negative feedback regulation from CHK2 to AR is not discussed here, for instance in the model in Figure 9. This aspect should be addressed, at least textually, to provide a complete picture of this crosstalk in PCa.

We have added text discussing the negative feedback model. In brief, we suggest that AR represses CHK2 to turn off the DDR, while CHK2 negatively regulates AR to dampen growth while damaged DNA is being repaired.

5) "Kinase-impaired CHK2 variants, including the K373E variant associated with 4.2% of PCa, blocked IR-induced CHK2-AR interactions". This is a strong statement considering the blot in Figure 3B doesn't appear representative of the quantitation shown – it seems the variants reduce binding but certainly don't seem to block it completely.

We agree that the CHK2 variants association with the AR is slightly enhanced with IR and not completely blocked. We have changed the wording to more accurately reflect the reduced binding in the presence of IR.

6) It seems IR increases the CHK2/AR interaction by increasing CHK2 phosphorylation and therefore its activity, which is an interesting finding. However, both the experimental design and Figure 4 (which describes this effect) are hard to understand. For example, the blots on the left vs. the respective quantitation on the right are confusing. The graphs look different (- or + radiation of the V5-CHK2 cells) but the blots look very similar. It would help if Figure 4 included a schematic of the experimental design.

We have completely revised the Far Western data reported in the paper, which is now shown in Figure 1. We now include a schematic of the technique, in Figure 1—figure supplement 1, and thank the reviewer for the suggestion. The data initially reported in Figures 4 and 5 have been removed for clarity and robustness.

7) Statistical analyses and quantifications should be provided for all figure and supplementary panels, when appropriate (e.g. Figure 5A, Figure 8C, Supplementary Figure 2, Figure 6—figure supplement 1).

We have now included a description of the statistical analysis for each figure in the legend.

8) Did the authors check whether CHK2 directly phosphorylates AR, and whether this phosphorylation is required for their binding?

We have examined whether CHK2 phosphorylates the AR on select sites, as AR phosphorylation has long been an interest of my laboratory. We have spent considerable effort in the intervening time since first submitting our manuscript (and in prior years) in an effort to answer this question. However, our experiments have not yet yielded robust reportable data.

9) Although the authors show that CHK2 binding with AR peaks at 1 hour, it would be ideal if the others would have tested additional early time points (such as 30 min, 45 min, 1.5h) to exactly determine the peaking of signal. Are these data available?

The data demonstrating CHK2 and AR peak binding at 1 hour was with endogenous proteins in LNCaP cells. To address the question raised by the reviewers we performed co-IP experiments at earlier time points following IR in LNCaP. In the process of conducting these experiments, we learned that the endogenous co-IP does not always work for reasons that are unclear, despite considerable effort on our part to determine reliable experimental conditions for a robust co-IP of endogenous CHK2 and AR. In contrast, the co-IP of exogenous proteins is robust. Therefore, we attempted to use the co-IP of exogenous CHK2 and AR to refine the kinetics of the interaction. However, exogenous expression of CHK2 leads to basal CHK2 T68 phosphorylation making it more difficult to define the kinetics of CHK2 – AR co-IP since the interaction is dependent on CHK2 T68 phosphorylation that is induced by IR. We could show data summarizing multiple replicates of the endogenous co-IP of CHK2 and AR, however, we feel this would be disingenuous without having defined the conditions to reliably perform the endogenous protein co-IP.

10) As ARS81 is phosphorylated by CDK1, does CDK1 inhibition (using small molecule inhibitors) impact IR induced association of CHK2 and AR?

We thank the reviewers for raising this interesting question. We have one replicate showing that inhibition of CDK1 with RO3306 reduces CHK2-AR interaction by co-IP. One challenge with this type of experiment is that RO3306 treatment results in complete cell cycle arrest at the G2/M transition. This is an inherent issue with all experiments that pharmacologically inhibit cell cycle regulators – it is difficult to separate the observations due to inhibition of the protein of interest from the fundamental alteration in the cell cycle that could indirectly effect the readout. For that reason, we did not pursue this approach further. The other challenge is that while CDK1 does phosphorylate the AR on S81, our data demonstrates that CDK9 is the predominate S81 kinase (PMID: 20980437). This raises the larger issue that the context of CDK1 phosphorylation of S81 may be functionally different then the CDK9 phosphorylation of S81 that happens during transcription of AR target genes; this question cannot be adequately addressed within the scope of this manuscript.

When considering that CHK2 may not be phosphorylating the AR (see response to concern # 8), we asked the fundamental question of whether CHK2 facilitated CDK1 – AR association, with subsequent plans to examine AR pS81. We used CHK2 knockdown to determine the impact on the co-IP of CDK1 and AR. However, when cells were transduced with CHK2 shRNAs, the co-IP experiments of endogenous CDK1 and AR were not consistent, similar to the lack of consistency of endogenous CHK2 – AR co-IPs (see response to concern #9). These technical challenges have limited our ability to further explore this interesting questions at this time.

11) Please check whether the ERK1/2 control blots in Figure 6B (C42) vs. 6C (22Rv1) have been accidentally duplicated.

Thank you for bringing this to attention. We have ensured that the correct corresponding blots are now shown.